# Exposure–Response Relationship and Doubling Risk Doses—A Systematic Review of Occupational Workload and Osteoarthritis of the Hip

**DOI:** 10.3390/ijerph16193681

**Published:** 2019-09-30

**Authors:** Yi Sun, Annette Nold, Ulrich Glitsch, Frank Bochmann

**Affiliations:** 1Unit Applied Epidemiology, Institute for Occupational Safety and Health of the German Social Accident Insurance, 53757 Sankt Augustin, Germany; annette.nold@dguv.de (A.N.); frank.bochmann@dguv.de (F.B.); 2Unit Musculoskeletal Workload, Institute for Occupational Safety and Health of the German Social Accident Insurance, 53757 Sankt Augustin, Germany; ulrich.glitsch@dguv.de

**Keywords:** osteoarthritis of the hip, coxarthrosis, exposure–response relationship, dose–response relationship, workload, occupational risk, risk quantification, doubling risk dose, systematic review

## Abstract

In this review, we critically evaluated the evidence of exposure–response relationships between occupational workload and the risk of hip osteoarthritis. The existing evidence was evaluated in order for us to extrapolate possible doubling risk doses for hip osteoarthritis. Comprehensive searches for epidemiological studies of hip osteoarthritis and occupational workload were performed in literature databases (PubMed, EMBASE, Cochrane Work and Google Scholar) and recent reviews up to February 2019. In total, 85 papers met the preliminary inclusion criteria, and 10 studies indicating an exposure-response relationship between occupational workload and hip osteoarthritis were identified. All studies were assessed on the basis of their study design, defined quality scores and relevant confounders considered. An exposure–response relationship between heavy lifting and the risk of hip osteoarthritis is consistently observed among the male populations but not among the female populations. We quantified the doubling risk doses in two studies in which both an exposure–response relationship and cumulative exposure doses were stated. These two studies provided the highest quality level of all studies published to date. The estimated doubling risk doses in these two studies lie between 14,761 and 18,550 tons (daily lifting 2.2–2.8 tons, 220 days/year for 30 years). These results can be used for workplace interventions to prevent hip osteoarthritis.

## 1. Introduction

Osteoarthritis (OA), a degenerative disorder of the joint cartilage and its underlying bone, is the most prevalent joint disease. OA of the hip produces significant morbidity and is a major public health problem throughout the world [1,2]. Clinical and epidemiological studies of hip OA have revealed a series of etiological factors including both localized factors (such as malformations or joint injuries) and systemic factors (such as obesity, race, sex, age and metabolic diseases) [2]. Owing to its relevance as a modifying risk factor for hip OA, occupational workload has long been a subject of epidemiological studies.

In 2012, we published a systematic review of the epidemiological evidence of workload (such as standing, sitting, walking, running, squatting, stair climbing and heavy lifting) being a risk factor for hip OA [3]. This review indicated that long-term heavy lifting and standing could increase the risk of hip OA. However, the quality difference between the underlying studies appears to play an important role in determining the size of the estimated effect. It appears that studies with low quality scores tended to report higher effect estimates than studies with higher quality scores [3,4]. We questioned whether this observation merely reflects a random phenomenon.

In order to evaluate whether study quality may systemically influence the size of effect estimates in epidemiological studies of hip OA, we updated our previous review in a recently published meta-analysis [5]. This former analysis demonstrated that study quality has a strong influence on the estimated size of effect. Well-designed, high-quality studies generally yielded lower effect estimates than studies of lower methodological quality [5]. Our meta-analysis indicated a strong need for more well-designed, high-quality studies in order to improve quantification of the causal factors of hip OA.

Hip OA is a chronic disease with multiple causes [2]. The main causes may be quite different from one case of hip OA to the next. In the general population, physical workload constitutes only a small fraction of the etiological causes of hip OA [2]. Therefore, quantification of the main cause of hip OA in individual cases is of major importance for management and prevention of work-related hip OA. The so-called “doubling risk” criterion is often used to quantify the primary cause of a disease. “Doubling risk (OR/RR ≥ 2)” means an attributable fraction of 50%: if a special workload leads to a doubling risk of hip OA, the probability that this workload is the main cause of hip OA is 50%. The probability of error due to identification of a physical workload as the main cause of hip OA will consequently not exceed 50%.

The “doubling risk dose” is an effect measure that has been widely used in genetic studies since the mid-1950s [6]. Recently, it has also been used to quantify the primary cause of certain musculoskeletal diseases, such as lumbar spine disorders [7] or hip OA [8]. Since previous analyses quantified the doubling risk dose of work-related hip OA by using external exposures [8], we questioned whether such an analysis can be properly interpreted.

In order to provide a reliable estimation of doubling risk doses for work-related hip OA, we conducted the present systematic review. The objective of this review is to extend our recently published meta-analyses [5]. Based on the library of our meta-analyses [5], we critically evaluate the evidence of dose-response relationships between heavy lifting and the risk of hip OA. The possible doubling risk doses of heavy lifting for hip OA will be quantified based upon the published dose–response relationship.

## 2. Materials and Methods

### 2.1. Systematic Literature Searches

Details of the literature searches conducted in this review have already been published elsewhere [5]. Briefly, based on the library established for our review published in 2012 (including publications up to 2010) [3], we conducted comprehensive searches in multiple databases, including PubMed, EMBASE, Cochrane Work and Google Scholar, for studies published between 1 January 2010 and 31 December 2017. Comparison with reference lists in 10 reviews published since 2010 completed our extended literature searches and revealed also additional studies published before 2010. Bibliographies and cross-referencing completed our literature searches.

For this review, we extended our literature searches to include studies published up to February 2019 in the same databases. Comparison with reference lists was extended to two additional reviews.

Details of the search strategy are described in Appendix A. To provide a clearer view of the available studies, we listed all studies from our updated library: studies considered for critical evaluation of exposure–response relationships between occupational workload and hip OA are described and referenced in the present publication. Studies excluded from the review are listed in Appendix A, together with the reasons for their exclusion and the references.

Exclusion criteria are primarily:Studies do not address the topic, or address non-idiopathic hip osteoarthritis;No occupational exposure data are available;Studies do not provide an exposure-response estimation.

We followed the Preferred Reporting Items for Systematic Reviews and Meta-Analyses (PRISMA) Statement for reporting all outcomes of our review (Appendix A).

### 2.2. Classification of Quality Level of Published Studies

The quality level of epidemiological evidence depends on numerous methodological issues, such as design, sample size, exposure assessment methods, relevant confounders considered, diagnostic criteria, statistical analysis methods used and so on. Each study identified in this paper was evaluated according to the quality of its case ascertainment (diagnosis criteria) and the exposure assessment methods used, as shown in Table 1 and Table 2 respectively. These classification schemas have already been reported in our recently published meta-analysis [5].

### 2.3. Statistical Analysis

In order to quantify the possible doubling risk doses of occupational workload for work-related hip OA, we performed the following analysis steps:Step 1: Estimation of the average (median) exposure values for each exposure category defined in published studiesIn order to extrapolate a steady course of the exposure–response relationship, we first had to estimate the median exposure values for each exposure category (which were not provided in the published studies). Since in all published studies the distribution of the cumulative workloads is distorted, we assumed that they follow a log-normal distribution. Based on an optimal fit to the described distribution of the study population in each exposure category, we simulated a steady distribution of occupational workload for each study and extrapolated the median exposure values for each exposure category.Step 2: Extrapolation of a steady course of the exposure–response relationshipThe estimated median exposure values of each exposure category were used to extrapolate a steady course of the exposure–response relationship for the published studies. We assume that the published exposure–response relationships follow a linear exponential function in which OR/RR = e^xß^, where x = cumulative exposure dose and ß = regression coefficient. This assumption serves as the basis of almost all statistical models used for exposure–response estimation (such as logistic regression, Cox model and Poisson regression analysis).Step 3: Quantification of the doubling risk dosesThe possible doubling risk doses can then be read off directly from the estimated exposure–response curve.

In order to evaluate the uncertainties of the estimated doubling risk doses, sensitivity analyses were carried out with the use of alternative median exposure values for each exposure category, in order to extrapolate the steady course of the exposure–response relationship and doubling risk doses.

All calculations were carried out for individual studies and pooled analysis of all studies by means of the Microsoft Excel 2016 (Microsoft Corporation, Redmond, WA, USA) and Stata 13 software packages (StataCorp, College Station, TX, USA).

## 3. Results

In total, 85 papers met the preliminary inclusion criteria (Figure 1). Following full text review by two independent researchers, 34 papers were selected for quality assessment in a sensitivity meta-analysis and meta-regression analysis of the evidence published [5]. A subset of 10 of the 34 studies yielded exposure-response relationships between occupational workload and the risk of hip OA [9,10,11,12,13,14,15,16,17,18]. No new studies relevant to our analysis were found in the updated searches for this review.

The PRISMA (Preferred Reporting Items for Systematic Reviews and Meta-Analyses) flow diagram shown in Figure 1 summarizes the process of the literature searches and selection of the studies.

A description of the study design, sample size, outcome assessment and exposure assessment methods used in the 10 studies is provided in Table 3.

One cross-sectional study [9], six population-based case-control studies [10,11,12,13,14,15] and three cohort studies [16,17,18] are available for quantitative exposure–response estimation. With the exception of one study [14], the sample sizes of the studies are relatively large. Seven studies attain the highest quality score of 3 for case ascertainment of hip OA [10,13,14,15,16,17,18] and nine studies achieve a good quality score of 3 for the exposure assessment methods used [9,10,11,12,13,14,15,17,18]. However, no study attains a quality score of 4–5 for exposure assessment. The important confounders of body-mass index (BMI) and prior injury were both considered in five of the 10 studies [9,10,13,16,18]. Overall, two population-based case-control studies [10,13] and one cohort study [18] have the highest methodological quality level (the studies concerned are all incidence studies with large sample size, highest quality score for case ascertainment, good quality score of exposure assessment methods, and complete control of important confounders).

Table 4 provides a detailed description of the exposure assessment methods used, available exposure data and effect estimates in the 10 studies.

All studies provided consistent findings of an exposure–response relationship between heavy lifting and the risk of hip OA among the male populations, but not among the female populations. In most of the studies, exposure assessment was based solely on personal interviews concerning lifetime work history and workload, conducted with the study populations themselves. In contrast, the cohort study by Rubak et al. [17,18] employed more objective methods for assessment of occupational exposures, involving pension data (for reconstruction of individual work history) and a standardized job-exposure matrix (JEM) for quantification of exposure levels.

Occupational workload was assessed incompletely [9,10,11,12] or not at all [14,15] in six studies. These studies failed to state the cumulative workload. The exposure–response relationships were stated only for exposure duration or frequency rather than in the form of cumulative exposure doses. Cumulative workload was stated in four studies [13,16,17,18]. In two of these, workload was considered not only for heavy lifting, but also for standing, walking, running, squatting and whole-body vibration [16,17]. Since the combined workloads were presented by the use of specific scoring indexes, their meanings are difficult to interpret and compare with other studies.

Quantification of doubling risk doses can consequently be performed only for two studies stating both the exposure–response relationship and cumulative exposure doses [13,18].


**Rubak et al. 2014 [18] (Incidence study; large sample size; case ascertainment score, 3; exposure assessment score, 3; adjustment of all four relevant confounders).**


Rubak carried out a nested case-control study among a cohort of the total Danish population employed full-time for ≥10 years between 1964 and 2006 [17]. The complete individual work history was reconstructed by the use of data from the Pension Fund Register. In addition, self-reported job titles in each part of the work history were collected and linked to a JEM with 689 occupational titles. The exposure values of the JEM are based on the judgement of five experts. Table 4 shows the exposure–response relationship published in this study between heavy lifting and the risk of hip OA. One ton-year was standardized in the study as lifting 1 ton per day for 1 year. Since 1 working year = 220 working days, 1 ton-year = 220 tons.

In this study, approximately 31% of exposed workers are in the lowest exposure category (0–2200 tons), 29% in the medium exposure category (2200–4400 tons) and 40% in the highest exposure category (4400–25,300 tons). This description clearly indicates a skewed distribution of occupational workload in this study. We assumed that the occupational workload in this study follows a log-normal distribution. With an optimal fit to the described proportions of the study population in each exposure category, we simulated a steady distribution of occupational workload for this study as shown in Figure 2. In the simulated data, 32% of exposed workers are in the lowest exposure category, 24% in the medium exposure category and 44% in the highest exposure category. Hardly any difference exists between the simulated and real data in terms of the proportions of the study population in each exposure category.

Based on the simulated distribution of occupational workload, we quantified the median exposure values for each occupational category. These are 1350 tons, 3750 tons and 9416 tons, corresponding to the 62th, 56th and 24th percentiles of the ranges of the low, medium and high exposure categories, respectively. We extrapolated the courses of the dose–response curve by using these values as shown in Figure 3. For sensitivity analyses, we additionally used the 20th and 30th percentiles of the range in the highest exposure category to represent its possible median workload. The estimated doubling risk doses are 13,136 tons, 14,761 tons and 17,198 tons, respectively, when 20th, 24th and 30th percentiles of the range of the highest exposure category are used to represent its median workload. Additional sensitivity analyses were also carried out by using alternative values of median workload in the lowest two exposure categories. Our analysis demonstrated that a 15% change in the median exposure values in the lowest two exposure categories has hardly any influence on the estimated values of doubling risk doses.


**Vingard et al. 1991 [13] (Incidence study; large sample size; case ascertainment score, 3; exposure assessment score, 3; adjustment of all four relevant confounders).**


Vingard carried out a population-based case-control study among Swedish men between 50 and 70 years of age. The occupational workload, including the lifetime number of kilograms lifted and the number of times heavy loads were lifted, was assessed in this study by telephone interview. In addition, biomechanical investigations of hip-joint load were performed under laboratory conditions. The dose–response relationship determined in this study between heavy lifting and the risk of hip OA is shown in Table 4.

In this study, 5% of exposed workers are in the lowest exposure category (0–137 tons), 47.5% in the medium exposure category (138–3006 tons) and 47.5% in the highest exposure category (3007–94,003 tons). This indicates that the distribution of occupational workload in this study is strongly skewed. With a log-normal assumption and an optimal fit to the described proportions of the study population in each exposure category, we simulated a steady distribution of occupational workload for this study as shown in Figure 2. In the simulated data, 6% of exposed workers are in the lowest exposure category, 46.6% in the medium exposure category and 47.4% in the highest exposure category. Hardly any difference exists between simulated and real data in terms of the proportions of the study population in each exposure category. Based on the simulated distribution of occupational workload, we quantified the median exposure values for each occupational category. These are 69 tons, 860 tons and 12,107 tons, corresponding to the 50th, 30th and 14th percentiles of the ranges of the low, medium and high exposure categories, respectively. The courses of the dose–response curve extrapolated by use of these values are presented in Figure 4. For sensitivity analyses, we additionally used the 10th and 20th percentiles of the range in the highest exposure category to represent its possible median workload. The estimated doubling risk doses are 14,100 tons, 18,550 tons and 25,130 tons, respectively, when 10th, 14th and 20th percentiles of the range of the highest exposure category were used to represent its median workload. Additional sensitivity analyses demonstrated that a 15% change in the values of the median workload in the lowest two exposure categories has hardly any influence on the estimated values of doubling risk doses.

In a pooled analysis of the two studies by Vingard et al. [13] and Rubak et al. [18], we quantify the course of the dose-response relationship between occupational heavy lifting and the risk of hip OA, as shown in Figure 5. The estimated doubling risk dose in the pooled analysis is 18,277 tons.

## 4. Discussion

In order to formulate a scientific basis for quantifying the main cause of work-related hip OA, we critically evaluated the evidence for dose-response relationships between occupational workload and the risk of hip OA in a systematic literature review. Among 34 relevant studies identified, we found 10 studies [9,10,11,12,13,14,15,16,17,18] stating a quantitative exposure-response relationship. Consistent findings of a positive exposure–response relationship can be observed among the male populations, but not among the female populations. This is consistent with our recently published meta-analysis [5] and a meta-analysis performed recently by Seidler et al. [8]. The inconsistency between the findings for the male and female populations may indicate a generally lower physical workload among the latter; there are indeed fewer women than men working in heavy lifting jobs, except in the healthcare professions. Conversely, female populations may experience high physical load of long duration in their “non-work-related” lives (e.g., pregnancy, childcare, etc.). The female populations may therefore experience higher cumulative physical load in their “non-work-related” lives than during their work.

Among the 10 studies providing an exposure–response relationship, interpretable cumulative exposures were stated in only two studies [13,18], which can be used to extrapolate the course of the dose–response relationship in order to quantify the possible doubling risk dose for work-related hip OA. Limited exposure data is a common problem in occupational epidemiological studies. A previous analysis used an external reference population to extrapolate the possible cumulative exposures in order to quantify the doubling risk doses of work-related hip OA [8]. We consider such extrapolation to be extremely unreliable. For instance, the two studies proving cumulative exposures in this analysis [13,18] are comparable with each other neither in their distribution, nor in the range of their exposure values. The use of one study to extrapolate from the other is extremely misleading. We therefore deliberately avoid making arbitrary assumptions to extrapolate the possible cumulative exposures for the eight studies in which cumulative exposure data were not stated. To ensure a high-quality estimation, we focus on the quantification of doubling risk dose only for the two studies stating both cumulative exposure doses and dose-response relationships [13,18].

The nested case-control study by Rubak et al. [18] provided the most objective measures of occupational exposures. The use of JEM based on experts’ judgement avoids certain individual recall biases and increases the overall validity of the exposure assessment. Since the JEM values used in this study [18] represent certain estimated average exposures for the job titles concerned, the individual exposure variability and overall range of exposure are expected to be much lower than that in the study by Vingard et al. [13]. It was demonstrated clearly in the two studies that the highest cumulative workload in the study by Vingard et al. (94,003 tons) is approximately 3.7 times higher than that in the study by Rubak et al. (25,300 tons). Interestingly, our simulated distributions of occupational workload indicate the same geometric means for the occupational workload in the two studies [13,18], which indicates that the two study populations share the same average occupational exposure. This is plausible, since both studies were carried out among the general populations in Nordic countries. The relatively comparable working conditions between the two study populations provide a sound methodological basis for a pooled analysis of their data [13,18].

In order to extrapolate a steady course for the dose–response relationship, we estimated the median exposure values for each exposure category defined in the two studies [13,18]. We assume that the cumulative workloads follow a log-normal distribution. The reason for this assumption is first that the distributions of occupational workload in both studies are extremely distorted. According to our experience, a naturally skewed distribution often follows a log-normal distribution. Second, we quantified the distribution of occupational workload in a previously published case-control study among a group from the German population [19]. Our analysis demonstrated that the distribution of cumulative heavy lifting within this population, comprising around 700 persons, exactly follows a log-normal distribution. With an optimal fit of the described proportions of the study population in each exposure category, we simulated the distribution of occupational exposures for these two studies [13,18] and quantified the median exposure values for each exposure category. We are aware that this analysis exhibits some uncertainties. Sensitivity analyses were therefore carried out during extrapolation of the course of the dose–response relationship. Our analyses indicate that the use of different median exposure doses in the highest exposure category has a strong influence on the estimated values of doubling risk doses. In contrast, 15% changes in median exposure in the lowest two exposure categories have less influence on the estimated values of doubling risk doses.

Before any conclusions can be drawn, certain limitations and strengths of our analysis must be carefully discussed:First, doubling risk doses of work-related hip OA can be quantified only in two studies that state both an exposure–response relationship and cumulative exposure doses. This limited number of studies may limit the generalization of our findings, and indicates some selections. However, the selection criteria are only the quality criteria presented in this paper. The two studies [13,18] selected in this analysis exhibit the highest methodological quality levels of all studies published to date (large sample size, highest quality score for case ascertainment, good quality score for exposure assessment methods, complete adjustment of relevant confounders). Were we to include more studies in the analysis, we would have to lower our quality requirements and introduce more bias or uncertainties into the analysis. This does not appear reasonable.Second, we assume that doubling risk doses exist at high exposure levels. However, we are unsure whether they can in fact be achieved in empirical studies. In the study by Rubak et al. [18] for example, the highest effect estimate reaches a level of OR = 1.35. We do not know whether the effect estimate would ever reach OR ≥ 2 at higher exposure levels. However, the conservative monotonic dose–response assumption may assist us in extrapolating a minimum possible value for the doubling risk dose. In fact, this value is more likely to be higher, if indeed it exists at all.Third, although cumulative exposure doses were stated in the two studies referenced in this analysis [13,18], we must make certain assumptions concerning the average (median) exposure values in each exposure category. Our analysis indicates that different assumptions of the exposure values in the highest exposure category have an influence upon the estimated values of doubling risk doses. This analysis nevertheless provides a range of possible values for doubling risk doses that can serve as an orientation in future for quantifying the possible main cause of work-related hip OA.Fourth, we would point out that workers subject to higher physical loads (such as farmers, transportation workers, construction workers, etc.) differ from the general population. They are often healthier and have a lower susceptibility to disease (typical healthy worker effect). Doubling risk doses quantified among the general populations in this analysis cannot therefore be extrapolated to all working groups, especially to workers subject to a higher physical workload. The values stated in this analysis are more likely to indicate underestimated minimum doubling risk doses for workers subject to high physical workloads.

In addition, the workload discussed here refers exclusively to the values of load handled without taking into account the different kinds of load handling. It should be noted that the compressive forces on the hip joints varies considerably by different ways of load handling (e.g., initial load heights, asymmetric handling of loads, duration of each lifting or carrying, etc.) [20]. These aspects need to be considered carefully in future studies in interpretations of the true burdens of physical workload, and in order to improve the common exposure assessments methods used to date.

## 5. Conclusions

Overall, the analyses covered by the present review demonstrate a clear dose–response relationship between occupational workload and the risk of hip osteoarthritis among the male populations, but not among the female populations. The quantification of doubling risk doses was carried out only in two high-quality studies stating both a dose–response relationship and cumulative exposure doses. The estimated doubling risk doses of heavy lifting for hip osteoarthritis are between 14,761 and 18,550 tons (daily lifting 2.2–2.8 tons, 220 days/year for 30 years). The findings from this analysis can help to quantify to what extent individual cases of hip osteoarthritis could be job-related. This can support efforts to improve workplace interventions in order to prevent hip osteoarthritis.

## Figures and Tables

**Figure 1 ijerph-16-03681-f001:**
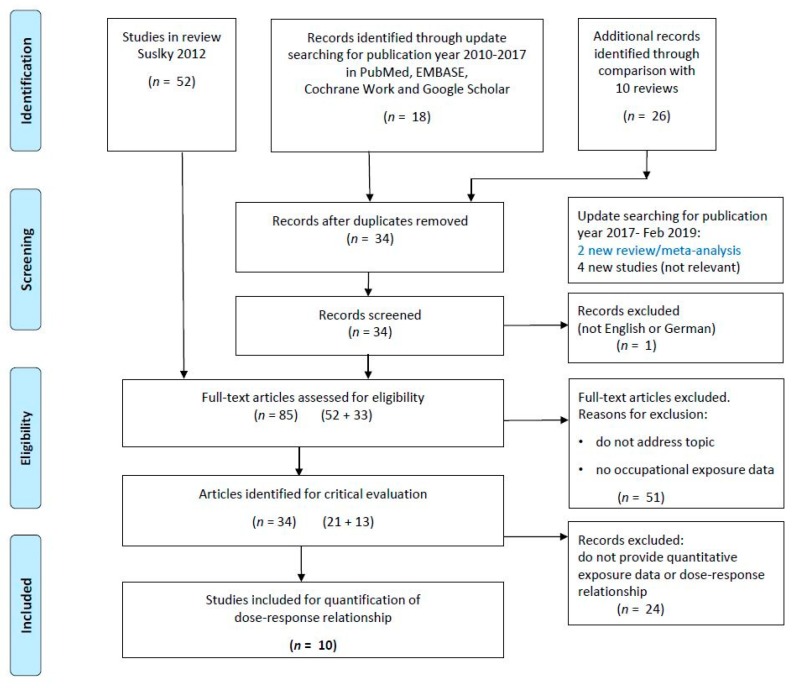
Preferred Reporting Items for Systematic Reviews and Meta-Analyses (PRISMA) flow diagram for the selection of literature for critical evaluation.

**Figure 2 ijerph-16-03681-f002:**
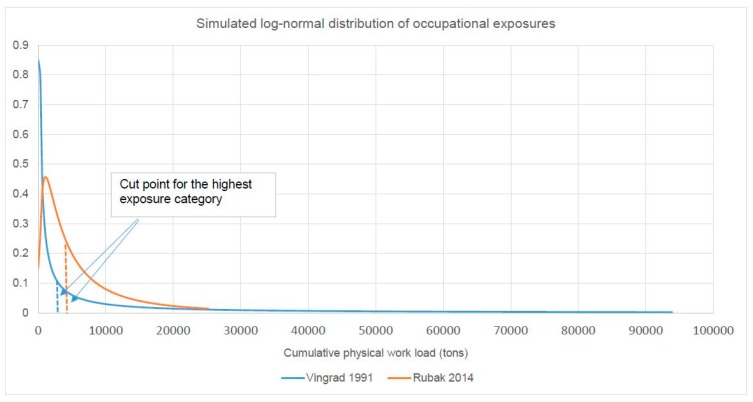
Simulated distributions of occupational workload in the studies by Rubak et al, [18] and Vingard et al. [13] based on published exposure data.

**Figure 3 ijerph-16-03681-f003:**
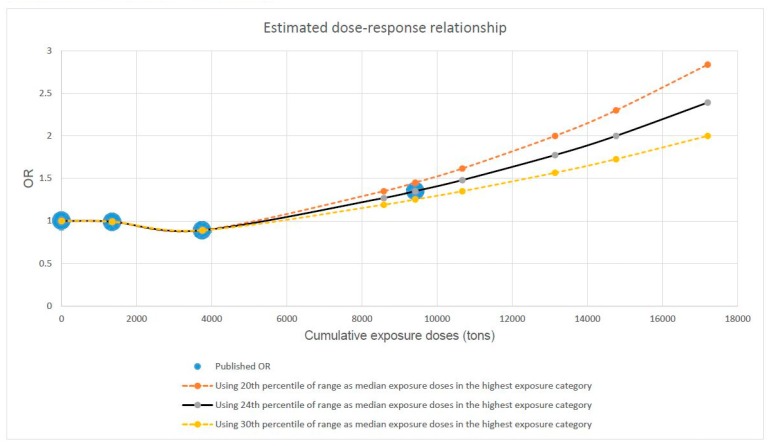
Extrapolated course of the dose–response relationship between heavy lifting and the risk of hip osteoarthritis based on findings published by Rubak et al. 2014 [18].

**Figure 4 ijerph-16-03681-f004:**
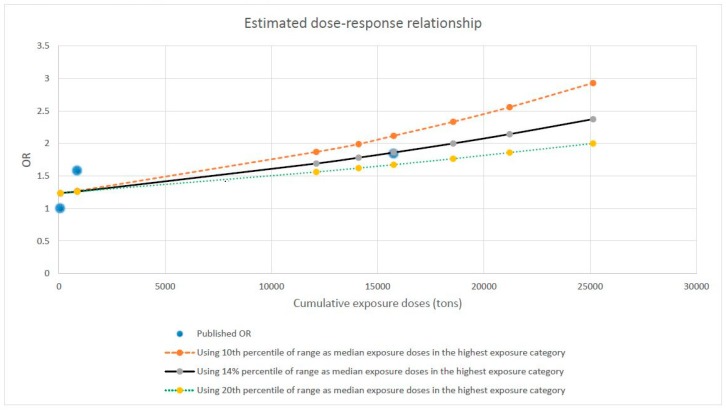
Extrapolated course of the dose–response relationship between heavy lifting and the risk of hip osteoarthritis based on findings published by Vingard et al. 1991 [13].

**Figure 5 ijerph-16-03681-f005:**
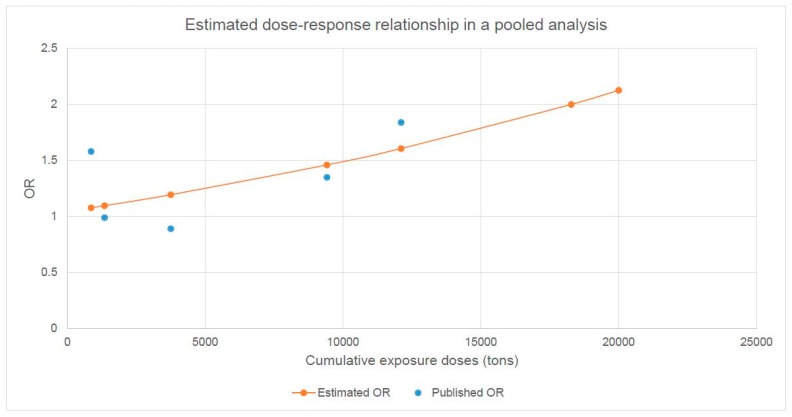
Extrapolated course of the dose–response relationship between heavy lifting and the risk of hip osteoarthritis in a pooled analysis of the data given by Vingard et al. 1991 [13] and Rubak et al. 2014 [18].

**Table 1 ijerph-16-03681-t001:** Definition of level of evidence for diagnostic evaluation of studies on osteoarthritis (OA) of the hip (score 1–3) [5].

Diagnosis Criteria	Diagnostic Quality Score * (Evidence Level)
Anamnesis/questionnaire: hip pain without clinical check	1
Hip pain and clinical reduction of movement without radiographic features or Radiographic features without clinical examination, without THR (total hip replacement)	2
Hip pain with clinical reduction of movement and clearly defined radiographic features (joint space narrowing or Kellgren–Lawrence-score grade 2 and above or comparable criteria) or diagnosis with indication for THR (total hip replacement)	3

* Score 1—low quality; score 3—high quality.

**Table 2 ijerph-16-03681-t002:** Definition of quality for exposure assessment of studies on osteoarthritis of the hip (score 1–5) [5].

Exposure Assessment	Exposure Quality Score *
Profession, job title, classification of occupation	1
Qualitative specification of exposure in different work activities (lifting, climbing stairs, sitting)	2
Quantitative specification of exposure in different work activities/physical strains with information on intensity (e.g., load weight steps) and duration	3
Quantitative specification of exposure (as above) with additional plausibility check (e.g., information on daily work output or special controls through video analysis)	4
Quantitative, measured exposure with quantitative assessment or modeling of hip joint strain	5

* Score 1—low quality; score 5—high quality.

**Table 3 ijerph-16-03681-t003:** Design and methodological quality of studies stating a dose–response relationship between heavy lifting and the risk of hip osteoarthritis.

Design	Study	Outcome Assessment	Study Population		Quality Score of		Confounders Controlled				Exposure Parameter Estimated
			Sample Size	Age (Years)	Exposure Assessment	Hip OA Ascertainment	Age	Sex	BMI	Prior Injury	
Cross- sectional	Kaila-Kangas 2011 [9]	Prevalence	6556	30–97	3	2	✓	✓	✓	✓	Lifting/carrying/pushing
Population based case-control	Coggon 1998 [10]	Incidence	611 cases 611 controls	45–91	3	3	✓	✓	✓	✓	Lifting
	Croft 1992 [11]	Prevalence	245 cases 294 controls	60–75	3	2	✓	✓			Lifting/moving
	Pope 2003 [12]	Prevalence	352 cases 3002 controls	18–85	3	1	✓	✓			Lifting/moving
	Vingard 1991 [13]	Incidence	239 cases 302 controls	50–70	3	3	✓	✓	✓	✓	Lifting
	Roach 1994 [14]	Prevalence	99 cases 233 controls	Mean 68	3	3	✓	✓	✓		Heavy work
	Vingard 1997 [15]	Prevalence	230 cases 273 controls	50–70	3	3	✓	✓	✓		Lifting
Cohort (including nested case-control)	Ratzlaff 2011 [16]	Incidence	2918	45–85	2	3	✓	✓	✓	✓	Peak hip joint force
	Rubak 2013 [17]	Incidence	1.9 million	31–71	3	3	✓	✓			Physical work
	Rubak 2014 [18]	Incidence	1776 cases 1776 controls	41–69	3	3	✓	✓	✓	✓	Lifting

**Table 4 ijerph-16-03681-t004:** Exposure data, exposure assessment methods and effect estimates of studies stating a dose–response relationship.

Design	Study	Exposure Assessment Methods	Exposure Values Assessed				OR/RR *		
			Weight Handled	Frequency (n)	Duration (years)	Exposure Doses	Male	Female	Both Sexes
Cross-sectional	Kaila-Kangas 2011 [9]	Personal interview for lifetime work history (99%)	>20 kg	≥10 times/day	0	Not available	1	1	1
					1–12		1.1 (0.4–3.2)	1.6 (0.7–3.5)	1.4 (0.7–2.6)
					13–24		2.2 (0.8–5.9)	3.8 (1.7–8.1)	2.8 (1.5–5.0)
					>24		2.3 (1.2–4.3)	1.2 (0.7–2.1)	1.8 (1.1–2.4)
Population based case-control	Coggon 1998 [10]	Personal interview for lifetime work history	≥25 kg	>10 times/week	0	Not available	1	1	1
					0.1–9.9		0.8 (0.4–1.7)	1.1 (0.6–1.7)	0.9 (0.6–1.4)
					10–19.9		1.5 (0.6–3.8)	1.4 (0.7–2.9)	1.2 (0.7–2.2)
					≥ 20		2.3 (1.3–4.4)	0.8 (0.4–1.5)	1.5 (1.0–2.3)
	Croft 1992 [11]	Personal interview for lifetime work history	>25.4 kg	Not provided	<1	Not available	1 (all cases)		
					1–19		0.9 (0.6–1.4)		
					≥20		1.2 (0.7–1.9)		
					<1		1 (severe cases)		
					1–19		1.2 (0.5–2.9)		
					≥20		2.5 (1.1.5.7)		
	Pope 2003 [12]	Personal interview for lifetime work history	>23 kg	Not provided	0	Not available			1
					1–12				1.02 (0.58–1.80)
					≥13				1.74 (1.06–2.86)
	Vingard 1991 [13]	Questionnaire for lifetime work history	Total load lifted	Per week		0–137 tons	1		
						138–3006 tons	1.58 (0.93–2.66)		
						3007–94,003 tons	1.84 (1.12–3.03)		
Population based case-control	Roach 1994 [14]	Questionnaire for the number of years of heavy work	Not provided	Not provided	0	Not available	1		
					15–24		2.2		
					25–34		3.0		
					>34		2.2		
	Vingard 1997 [15]	Questionnaire for lifetime work history	Not provided	0–20,328	Not provided	Not available		1.0	
				20,329–44,088				1.1 (0.7–1.7)	
				44,089–95,040				1.5 (0.9–2.5)	
Cohort (including nested case-control)	Ratzlaff 2011 [16]	Questionnaire for lifetime work and leisure time activities, estimation of lifetime CPFI **	Standing, running, squatting, carrying			1st quintile			1
						2nd quintile			1.11 (0.63–1.83)
						3rd quintile			1.3 (0.72–2.11)
						4th quintile			1.58 (0.86–2.52)
						5th quintile			1.80 (0.95–2.82)
	Rubak 2013 [17]	Complete work history by pension register, development of industry exposure matrix (IEM)	Lifting, walking, whole-body vibration			0 (point years)	1	1	
						>0 to <5	1.13 (0.98–1.31)	0.96 (0.8–1.06)	
						5 to <15	1.14 (1.00–1.31)	0.96 (0.87–1.05)	
						15 to <25	1.19 (1.04–1.36)	0.94 (0.85–1.04)	
						25 to <35	1.27 (1.11–1.48)	0.99 (0.88–1.10)	
						35–86	1.33 (1.17–1.53)	1.01 (0.88–1.16)	
Cohort (including nested case-control)	Rubak 2014 [18]	Self-reported lifetime job title, job-exposure matrix (JEM) for load lifted	Total load lifted	Per day		0 (ton-years)	1	1	
						>0 to <10	0.99 (0.75–1.30)	1.15 (0.87–1.53)	
						10 to <20	0.89 (0.67–1.17)	0.81 (0.61–1.09)	
						20–115/86	1.35 (1.05–1.74)	1.00 (0.72–1.41)	

* OR = odds ratio; RR = relative risk; ** CPFI: cumulative peak hip joint force index (% body weight).

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
