# Peer review of "Exposure–Response Relationship and Doubling Risk Doses—A Systematic Review of Occupational Workload and Osteoarthritis of the Hip"

_ijerph, 2019, doi:10.3390/ijerph16193681_

Round 1
Reviewer 1 Report
The submitted paper describes an interesting subject. It is quite well written, however, some minor changes are required. It is needed to extend Fig. 1 as it is invisible. There is such information: Supplementary Materials: The following are available online at www.mdpi.com/xxx: File S1: Literature search 325 strategy, Table S1: Excluded studies, reason for exclusion and references, Table S2: PRISMA checklist. It is not possible to find there a file.
In the tables the autohrs use '?" - what does it mean? it is discussed or not... please clarify its meaning and provided detailed information.
Author Response
Response to Reviewer 1 Comments
The submitted paper describes an interesting subject. It is quite well written, however, some minor changes are required.
It is needed to extend Fig. 1 as it is invisible. There is such information: Supplementary Materials: The following are available online at www.mdpi.com/xxx: File S1: Literature search 325 strategy, Table S1: Excluded studies, reason for exclusion and references, Table S2: PRISMA checklist. It is not possible to find there a file.
We are very sorry that Fig. 1 and some supplementary materials are not visible for the reviewer. All missing files were delivered to the IJERPH Editorial Office. So please contact them. The files are available from the Editorial Office.
In the tables the authors use '?" - what does it mean? it is discussed or not... please clarify its meaning and provided detailed information.
We agree with the reviewer’s comment and updated Table 4.

Reviewer 2 Report
This is a relevant paper regarding the relationship between occupational workload and hip osteoarthritis. In particular, the exposure-response relationship and a possible doubling risk dose is studied in order to emphasize the possible work-relatedness of the condition. In general the paper is well-written, I have been reading it with pleasure.
Comments:
Introduction
In some parts, the paper suffers from the reference to earlier papers/reviews by the same authors. The sketch of the background in the Introduction section is rather short and in my opinion not optimally informative. Maybe, the authors could give a little more information about the results of the earlier papers and what these results mean for the research question of this systematic review. The research question for this review can be formulated more clearly. I miss a definition or description of ‘occupational workload’. Also the purpose of looking at this risk doubling dose could be explained more comprehensively. In some countries, a PAF ≥ 50% is regarded as an important criterion for notifying or accepting the occupational cause of the diseases, sometimes even a criterion for workers compensation of an occupational disease.
When talking about doubling risk doses of occupational workload, about which risks are we talking about? Only lifting is mentioned in the Introduction. Later, it gets a more clear. More information for the reader could be given in the Introduction.
Methods
Also the Methods section suffers a bit from the reference to the earlier reviews. No clear search strategy is given and the additional file S1 is rather limited. Why these search terms and not others? Inclusion criteria are not presented. Were any other inclusion criteria relevant, e.g. on study design? I can imagine that for answering the research question of this review primarily longitudinal studies with also data on cumulative exposure are (most) informative. This choice has, however, not been made.
The quality rating system is, with regard to diagnostic criteria and exposure assessment, in itself nicely described. In many systematic reviews today, it is preferred to use systems for assessing ‘risk of bias’, indicating e.g. selection bias, performance bias, misclassification bias, etc. leading to a judgement of ‘low’, probably low, probably high, high or not applicable risk of bias which can also be graphically presented in a heat map. The authors have chosen not to do so.
Results
Most of the results are nicely presented. Study 7 and 12 are the key studies for answering the question of this review. Given the similarity of these studies, was it not possible to synthesize the results of these studies in a (limited) meta-analysis? If not, please give the reasons for refraining from this.
Discussion
The first part of the discussion section does not reflect the answer to the research question and/or the main findings. I also miss a more thoroughly discussion of the findings of this review with findings by other authors of (systematic) reviews or original studies on this topic. Also, I miss some recommendations for future research, e.g. research on the reasons why in male workers an association was found and not in female workers.
Author Response
Response to Reviewer 2 Comments
This is a relevant paper regarding the relationship between occupational workload and hip osteoarthritis. In particular, the exposure-response relationship and a possible doubling risk dose is studied in order to emphasize the possible work-relatedness of the condition. In general the paper is well-written, I have been reading it with pleasure.
Comments:
Introduction
In some parts, the paper suffers from the reference to earlier papers/reviews by the same authors. The sketch of the background in the Introduction section is rather short and in my opinion not optimally informative. Maybe, the authors could give a little more information about the results of the earlier papers and what these results mean for the research question of this systematic review. The research question for this review can be formulated more clearly.
We agree with the reviewer’s comment and provide more background information about our previous analyses (line 39 – 71).
I miss a definition or description of ‘occupational workload’.
We agree with the reviewer’s comment and added the description of ‘occupational workload’ (line 39-41).
Also the purpose of looking at this risk doubling dose could be explained more comprehensively. In some countries, a PAF ≥ 50% is regarded as an important criterion for notifying or accepting the occupational cause of the diseases, sometimes even a criterion for workers compensation of an occupational disease.
We agree with the reviewer’s comment and are aware that the issue of occupational disease is extremely important. However, the legal definition of occupational disease requires multidimensional considerations besides the scientific evidences. In this paper, we prefer to focus our analysis only on the scientific issue of work-relatedness of hip OA. We hope that the evidences presented in this paper can be considered in future in a multidimensional discussion of legal occupational disease for work-related hip OA.
When talking about doubling risk doses of occupational workload, about which risks are we talking about? Only lifting is mentioned in the Introduction. Later, it gets a more clear. More information for the reader could be given in the Introduction.
We agree with review’s comment and added more description about our previous analyses, especially regarding the relevant occupational risk factors for hip OA and the importance of methodological quality in determining the risk of hip OA. For more clarification, we updated the objective of this review by using “heavy lifting” instead of “occupational workload” (line 69 – 70).
Methods
Also the Methods section suffers a bit from the reference to the earlier reviews. No clear search strategy is given and the additional file S1 is rather limited. Why these search terms and not others?
The search strategy is provided in “Methods”, in Fig. 1 and supplement File S1. We updated Fig. 1 and File S 1 and described in detail our strategy in File S1. In our previous meta-analysis [5], comprehensive test searches were done for improving precision and recall.
Our previous review [3] indicated that only long-term ‘heavy lifting/carrying’ and ‘standing’ are relevant occupational risk factors for work related hip OA. ‘Heavy lifting/carrying’ and ‘farming’ are the most commonly used exposure measures in published studies [5]. In contrast, there are only limited number of studies addressing other types of physical workload [5]. Furthermore, heavy lifting/carrying is the only quantifiable measure for assessing doubling risk doses of hip OA. Therefore, we focus our search terms mainly on “heavy lifting”, “carrying” and “farming”. For more clarification, we updated the objective of this review by using “heavy lifting” instead of “occupational workload” (Introduction: line 69-70).
To ensure the completeness of literature search, we compared our review with recently published 12 reviews/meta-analysis on the same topic (s. list in S1-File). We did not find further relevant studies for this review.
Inclusion criteria are not presented. Were any other inclusion criteria relevant, e.g. on study design? I can imagine that for answering the research question of this review primarily longitudinal studies with also data on cumulative exposure are (most) informative. This choice has, however, not been made.
Since the purpose of this review is not only the quantification of doubling risk doses, but also the assessment of exposure-response-relationship, we included all types of designs providing exposure-response-estimation (including cross-sectional studies and those that do not provide cumulative exposures). We want to know, if there is a consistent findings of an exposure-response-relationship despite of the study designs and methodological qualities of the published studies. However, the quantification of doubling risk doses was limited only among the high quality studies providing cumulative exposures. In short, all studies are included except those excluded from this review due to the exclusion criteria.
The quality rating system is, with regard to diagnostic criteria and exposure assessment, in itself nicely described. In many systematic reviews today, it is preferred to use systems for assessing ‘risk of bias’, indicating e.g. selection bias, performance bias, misclassification bias, etc. leading to a judgement of ‘low’, probably low, probably high, high or not applicable risk of bias which can also be graphically presented in a heat map. The authors have chosen not to do so.
I think, depending on the preference of the authors, different quality-rating system can be used. I personally believe that ‘assessing risk of bias’ can be more effectively used for experimental designs with a clear outcome definition, effect definition and quality standard. In contrast, in observational studies, especially occupational epidemiological studies, there is often a lack of gold standard in case ascertainment, effect definition (including the use of reference population) and so on. Therefore, ‘assessing risk of bias’ can sometimes not be applied with satisfaction. In this analysis, we prefer to use ‘quality criteria’ instead of ‘risk of bias’, since the purpose of this analysis is to find the highest quality studies for quantifying doubling risk doses (although these quality studies still suffer from some risk of bias). We believe that the quality-rating system used in this paper is more effective.
Results
Most of the results are nicely presented. Study 7 and 12 are the key studies for answering the question of this review. Given the similarity of these studies, was it not possible to synthesize the results of these studies in a (limited) meta-analysis? If not, please give the reasons for refraining from this.
We agree with the reviewer’s comments, and pooled the two studies in an exposure-response estimation as show in fig.5. Doubling risk dose was also quantified based on the pooled analysis.
Discussion
The first part of the discussion section does not reflect the answer to the research question and/or the main findings.
We agree with the reviewer’s comment and corrected the section (delete the section from this paper.)
I also miss a more thoroughly discussion of the findings of this review with findings by other authors of (systematic) reviews or original studies on this topic.
We agree with the reviewer’s comment, and added discussions about studies by other authors on the same topic (Introduction: line 61 – 65; Discussion: line 294-296, 305-314)
Also, I miss some recommendations for future research, e.g. research on the reasons why in male workers an association was found and not in female workers.
We agree with the reviewer’s comment, and added some discussions about the reasons of gender differences and give some recommendations for future researches (line 384-386)
With pleasure I have read the paper entitled ‘Exposure-response relationship and doubling risk doses – a systematic review of occupational workload and osteoarthritis of the hip’.
This paper addresses a relevant topic in occupational medicine regarding the assessment whether work might be seen as the main cause for hip OA given that a doubling dose is often related to a 50% attributable fraction dependent on the comparison population.
We thank the reviewer for this comment.

Reviewer 3 Report
With pleasure I have read the paper entitled ‘Exposure-response relationship and doubling risk doses – a systematic review of occupational workload and osteoarthritis of the hip’.
This paper addresses a relevant topic in occupational medicine regarding the assessment whether work might be seen as the main cause for hip OA given that a doubling dose is often related to a 50% attributable fraction dependent on the comparison population.
Hopefully my comments are of use.
Title: ok
Abstract:
Despite the filled in PRISMA statement I think that essential information is missing like data sources used; study eligibility criteria, search date, number of studies found and included, number of participants, risk factors; study appraisal and synthesis methods, data on exposure-response relationship and doubling risk doses.
Please add what is the metric is for ’14,761 and 18,550 tons’, lifting or carrying per day for how many years?
Introduction:
Although I am in favor of short introductions I would appreciate a little more back ground data regarding the present knowledge on this topic. For instance, what are work and non-work-related risk factors for hip OA, what doses are seen as being ‘at risk’, what do we know from similar studies on this topic regarding doubling doses for instance for knee OA or shoulder pain, what criteria are used in guidelines for the assessment of hip OA as an occupational disease? Do studies indeed report doubling doses for work-related risk factors and hip OA?
I was surprised that the paper on a the same topic by Seidler and colleagues - Seidler et al. BMC Musculoskeletal Disorders (2018) 19:182, https://doi.org/10.1186/s12891-018-2085-8 - was not mentioned in the introduction: Dose-response relationship between cumulative physical workload and osteoarthritis of the hip – a meta-analysis applying an external reference population for exposure assignment. What does your paper add to their conclusions?
Methods:
How sensitive are your four search terms for exposure assessment : lifting, carrying, farmer?, farming. Did you validate these terms based on preselected studies that fulfill your inclusion criteria?
Thanks for the file with the papers that you have excluded including the reason.
I would have expected that only studies would have been included that are based on prospective cohort studies and that have corrected for established risk factors like sex, age, BMI, and trauma and used cumulative exposures. Why are all studies a priori eligible given your research question? (Of course your two remaining studies 7 and 12 nearly comply for all these criteria.)
Same goes for criteria for the assessment of exposure-criteria and reference populations. Why are these not taking into account a priori?
What do you mean by ‘distorted’ in statistical analyses 2.3: ‘…all published studies the distribution of the cumulative workloads is distorted.’
Why do you expect that for the exposure categories the distribution is lognormal within a given category and not just normal. If only one or two categories are given I can understand but is this also the case if three or more categories are specified in a study?
I like the way you have evaluated the diagnosis criteria and exposure assessment. Why does your quality assessment only takes into account these two factors despite your own remark on ‘…design, sample size, exposure assessment methods, relevant confounders considered, diagnostic criteria, statistical analysis methods used and so on.’
How do you handle studies that do not provide sufficient data for ‘Based on an optimal fit to the described distribution of the study population in each exposure category, we simulated a steady distribution of occupational workload for each study and extrapolated the median exposure values for each exposure category. ’?
How do you beforehand group data based on different exposure categories?
Regarding your step 3: How do you handle studies that did not find a doubling risk for their highest exposure category? Do you feel that it is scientifically appropriate to the estimate the doubling risk even though the study in question did not find this doubling risk based on their highest exposure category? How practical relevant are then these exposure criteria?
Results
Informative tables and figures!
Is it also possible to provide 95% confidence intervals for figures 3 and 4?
Why did you not pool the data for studies 7 and 12?
Discussion
Please start your discussion with the main findings
The second paragraph is more relevant for the introduction and can be left out of the discussion, I think.
At the moment the discussion merely reflects a detailed discussion of your own results instead of discussing the main findings of your paper in light of other studies on this topic, for instance the already mentioned Seidler paper or the paper by Verbeek on knee OA: https://www.ncbi.nlm.nih.gov/pmc/articles/PMC5447410/. See also the PRISMA statement: Provide a general interpretation of the results in the context of other evidence, and implications for future research.
What are the clinical implications of your findings?
Methodological limitations and strengths are well described regarding the included studies
References
See remarks in the introduction and discussion
No additional comments regarding figures and tables
Author Response
Response to Reviewer 3 Comments
Hopefully my comments are of use.
Title: ok
Abstract:
Despite the filled in PRISMA statement I think that essential information is missing like data sources used; study eligibility criteria, search date, number of studies found and included, number of participants, risk factors; study appraisal and synthesis methods, data on exposure-response relationship and doubling risk doses.
We agree with the reviewer’s comment and added more information to the abstract of this paper (such as data sources used, search date, number of studies found and included, data on exposure-response relationship and doubling risk doses). According to the standard of the journal, abstract was limited only to about 200 words maximum. Therefore, only key information of this manuscript can be provided.
Please add what is the metric is for ’14,761 and 18,550 tons’, lifting or carrying per day for how many years?
We agree with the reviewer’s comment and added the metric for ’14,761 and 18,550 tons’
Introduction:
Although I am in favor of short introductions I would appreciate a little more back ground data regarding the present knowledge on this topic. For instance, what are work and non-work-related risk factors for hip OA, what doses are seen as being ‘at risk’, what do we know from similar studies on this topic regarding doubling doses for instance for knee OA or shoulder pain, what criteria are used in guidelines for the assessment of hip OA as an occupational disease? Do studies indeed report doubling doses for work-related risk factors and hip OA?
We agree with reviewer’s comment and added more background knowledge about work and non-work-related risk factors for hip OA (line 32-41). We add also data of similar studies reporting exposure-response relationship and doubling risk doses of various conditions, including work-related hip OA (line 61-65). Since Hip OA is currently not a legal occupational disease in Germany, we cannot discuss the current criteria or guidelines for assessing it.
I was surprised that the paper on a the same topic by Seidler and colleagues - Seidler et al. BMC Musculoskeletal Disorders (2018) 19:182, https://doi.org/10.1186/s12891-018-2085-8 - was not mentioned in the introduction: Dose-response relationship between cumulative physical workload and osteoarthritis of the hip – a meta-analysis applying an external reference population for exposure assignment. What does your paper add to their conclusions?
We agree with reviewer’s comment and introduced the study by Seidler et al. 2018 in the introduction of this manuscript (line 61-65).
Methods:
How sensitive are your four search terms for exposure assessment: lifting, carrying, farmer?, farming. Did you validate these terms based on preselected studies that fulfill your inclusion criteria?
Our previous review [3] indicated that only long-term ‘heavy lifting/carrying’ and ‘standing’ are relevant occupational risk factors for work related hip OA. ‘Heavy lifting/carrying’ and ‘farming’ are the most commonly used exposure measures in published studies [5]. In contrast, there are only limited number of studies addressing other types of physical workload [5]. Furthermore, heavy lifting/carrying is the only quantifiable measure for assessing doubling risk doses of hip OA. Therefore, we focus our search terms mainly on “heavy lifting”, “carrying” and “farming”. For more clarification, we updated the objective of this review by using “heavy lifting” instead of “occupational workload” (Introduction: line 69-70).
To ensure the completeness of literature search, we compared our review with recently published 12 reviews/meta-analysis on the same topic (s. list in S1-File). We did not find further relevant studies for this review.
Thanks for the file with the papers that you have excluded including the reason.
I would have expected that only studies would have been included that are based on prospective cohort studies and that have corrected for established risk factors like sex, age, BMI, and trauma and used cumulative exposures. Why are all studies a priori eligible given your research question? (Of course your two remaining studies 7 and 12 nearly comply for all these criteria.)
We agree with the reviewer’s comment that cohort studies (or incidence studies) provide more convincing evidence than prevalence studies. Since the purpose of this review is not only the quantification of doubling risk doses, but also the assessment of exposure-response-relationship, we include all type of studies providing exposure-response-estimation. We want to know whether there is a consistent finding of an exposure-response-relationship despite of the different designs and methodological qualities. For the quantification of doubling risk doses, we limited the analyses only among the two high quality studies providing cumulative exposures.
Same goes for criteria for the assessment of exposure-criteria and reference populations. Why are these not taking into account a priori?
Due to the same reason mentioned above, we did not set priori condition for exposure assessment methods and reference population used in assessing exposure-response-relationship. However, for the quantification of doubling risk doses, we include only high quality studies.
What do you mean by ‘distorted’ in statistical analyses 2.3: ‘…all published studies the distribution of the cumulative workloads is distorted.’ Why do you expect that for the exposure categories the distribution is lognormal within a given category and not just normal. If only one or two categories are given I can understand but is this also the case if three or more categories are specified in a study?
I am afraid that there is a misunderstanding. We stated only that the distribution of cumulative workload of the whole study population is distorted. We expected also a lognormal distribution of occupational workload of the whole study population (all categories together). For each exposure category, no assumption can be made (they can be normal, lognormal or some others).
I like the way you have evaluated the diagnosis criteria and exposure assessment. Why does your quality assessment only takes into account these two factors despite your own remark on ‘…design, sample size, exposure assessment methods, relevant confounders considered, diagnostic criteria, statistical analysis methods used and so on.’
We agree with reviewer’s comment that the quality assessment can be extended to other methodological aspects such as design, sample size and relevant confounders and so on. Since there is only two studies available for quantifying doubling risk doses, we did not make more efforts to extend the quality criteria to other methodological aspects.
How do you handle studies that do not provide sufficient data for ‘Based on an optimal fit to the described distribution of the study population in each exposure category, we simulated a steady distribution of occupational workload for each study and extrapolated the median exposure values for each exposure category. ’?
Sufficient data means 1) the study provides cumulative exposure values for each exposure category 2) the study provides the proportion of study population in each exposure category. Without these two data, we cannot simulate the distribution of exposure for that study. Fortunately, the two key studies provide such information.
How do you beforehand group data based on different exposure categories?
There is no need to beforehand group data. With the conventional mathematical formula for lognormal distribution, you can simulate any form of lognormal distribution. By changing the parameters (geometric mean and SD) of the mathematical formula, you can find the best fit to the data given in the study.
Regarding your step 3: How do you handle studies that did not find a doubling risk for their highest exposure category? Do you feel that it is scientifically appropriate to the estimate the doubling risk even though the study in question did not find this doubling risk based on their highest exposure category? How practical relevant are then these exposure criteria?
This is a very good and important question. We have discussed this issue in the section ‘Discussion’ (line 359 – 364)
Results
Informative tables and figures!
Is it also possible to provide 95% confidence intervals for figures 3 and 4?
We do not think, it is a good idea to present 95% CI in this analysis. The overlapping confidence interval will make figure 3 and 4 more confusing. I think, sensitivity analysis gives more clear indication about the uncertainties of this analysis than confidence interval.
Why did you not pool the data for studies 7 and 12?
We agree with the reviewer’s comments and pooled the two studies in an exposure-response estimation as show in fig.5. Doubling risk dose was quantified based on the pooled analysis.
Discussion
Please start your discussion with the main findings. The second paragraph is more relevant for the introduction and can be left out of the discussion, I think.
We agree with the reviewer’s comment, and removed the 1st and 2nd paragraph.
At the moment the discussion merely reflects a detailed discussion of your own results instead of discussing the main findings of your paper in light of other studies on this topic, for instance the already mentioned Seidler paper or the paper by Verbeek on knee OA: https://www.ncbi.nlm.nih.gov/pmc/articles/PMC5447410/. See also the PRISMA statement: Provide a general interpretation of the results in the context of other evidence, and implications for future research.
We agree with the reviewer’s comment and discussed the findings by Seidler et al. (line 296, line 305-314). This is the only study we found addressing the same topic.
What are the clinical implications of your findings?
Our review cannot deliver any clinical implications, but the general implications of our findings was added in the last sentence of this paper.
Methodological limitations and strengths are well described regarding the included studies
References
See remarks in the introduction and discussion
No additional comments regarding figures and tables

Round 2
Reviewer 3 Report
all my comments were addressed.